# Matched Unrelated Donor Hematopoietic Cell Transplantation: Increased Usage and Improvements in Clinical Outcomes in Canada

**DOI:** 10.3390/curroncol32010010

**Published:** 2024-12-27

**Authors:** Matthew D. Seftel, Grace Musto, David Allan, Oliver Bucher, Kevin Hay, Ivan Pasic, Tony Truong, Kristjan Paulson

**Affiliations:** 1Stem Cells, Canadian Blood Services, Ottawa, ON K1Z 7M3, Canada; david.allan@blood.ca; 2Department of Epidemiology, CancerCare Manitoba, Winnipeg, MB R3A 1M5, Canadaobucher@cancercare.mb.ca (O.B.); 3Division of Hematology, Department of Medicine, University of Calgary, Calgary, AB T2N 1N4, Canada; kevin.hay@ucalgary.ca; 4Hans Messner Allogeneic Blood and Marrow Transplantation Program, Division of Medical Oncology and Hematology, Princess Margaret Cancer Centre, University Health Network, Toronto, ON M5G 2M9, Canada; ivan.pasic@uhn.ca; 5Division of Pediatric Hematology/Oncology, Department of Pediatrics, University of Calgary, Calgary, AB T3B 6A8, Canada; tony.truong@albertahealthservices.ca; 6CancerCare Manitoba, University of Manitoba, Winnipeg, MB R3E 0V9, Canada; kpaulson@cancercare.mb.ca

**Keywords:** hematopoietic cell transplantation, donors, real world evidence

## Abstract

In allogeneic hematopoietic cell transplantation (HCT), a minority of patients have access to a suitable human leukocyte antigen (HLA)-matched related donor (MRD). To fill this gap, matched unrelated donors (MUDs) are an increasingly selected donor source. Usage and outcomes after MUD HCT for Canada are not described. We investigated temporal trends in MUD compared to MRD HCT from 2000 to 2019 using data reported to the Cell Therapy and Transplant Canada (CTTC) Registry. Of 7571 first allogeneic HCTs between 2000 and 2019, the proportion of MUD HCTs rose from 35.1% to 56.3% in the early (2000–2009) and later (2010–2019) eras, respectively. Comparing the two donor sources, the 5-year overall survival (OS) after MUD HCT for patients with malignant diseases was inferior to MRD HCT in the early era (*p* < 0.001). However, in the later era, OS was comparable for the two donor sources (*p* = 0.969). For patients with non-malignant diseases, the 5-year OS after MUD HCT was inferior to MRD in the early era (*p* < 0.001), but in the later era, the 5-year OS was similar between the two donor sources (*p* = 0.209). Improvements in OS after MUD HCT were accompanied by corresponding reductions in the 2-year non-relapse mortality after MUD HCT. We conclude that MUDs are the most common donor source in Canada, and key clinical outcomes after MUD have improved over time.

## 1. Introduction

Allogeneic hematopoietic cell transplantation (HCT) is an established procedure for many life-threatening malignant and non-malignant hematopoietic conditions. Since the inception of allogeneic HCT, the optimal donor has been a human leukocyte antigen (HLA)-matched related donor (MRD), typically a sibling [1]. However, most patients do not have an available MRD, especially in the case of small family sizes [2]. To address this gap in donor availability, increasing numbers of HLA-matched unrelated donor (MUD) HCT procedures have occurred since the 1990s [3].

MUD HCT has historically been associated with higher transplant-related morbidity and mortality in the recipient. Consequently, MUDs have represented a second and less ideal donor choice compared to MRD [1]. However, improved laboratory methods to measure HLA polymorphisms have led to more accurate, high-resolution matching between recipients and unrelated donors. In addition, advances in supportive care have improved early outcomes after HCT, particularly with respect to the prevention and management of opportunistic infections and graft-versus-host disease (GVHD) [4].

Although observational data suggest that outcomes after MUD use may be comparable to those of MRD use for patients with malignant diseases [3], these results have not been confirmed in a randomized controlled trial (RCT). Consequently, there remains residual uncertainty about the equivalence of MUDs compared to MRDs as donor sources for allogeneic HCT, especially for patients with malignant diseases, where GVHD is associated with the graft-versus-malignancy effect, hence offering a potential therapeutic advantage. Observational data further suggest that the use of younger MUDs may reduce relapse compared to older MRDs in patients with AML and ALL [5,6,7].

The Cell Therapy and Transplant Canada Registry recently published trends and outcomes after HCT in Canada [8]. This observational study showed that overall numbers of allogeneic HCTs have increased between 2000 and 2019. Overall survival (OS) improved over this time for adult allogeneic HCT recipients with malignant diseases. OS also improved for pediatric HCT recipients with either malignant or non-malignant disease. However, it is unknown whether these observed temporal improvements in outcome are associated solely with the use of MRDs, or whether such benefits can be extended to MUD allografts. To assess whether outcomes after MUD HCT have improved in Canada, we further scrutinized data from the Cell Therapy and Transplant Canada (CTTC) Registry.

## 2. Methods

The current analysis is an extension of the retrospective cohort study of the first adult and pediatric HCTs performed in Canada, which used data reported to the CTTC Registry [7]. HLA-matched transplants were defined as either 8/8 or 10/10 allele-matched. Outcomes were stratified into “early” (2000–2009) and “late” (2010–2019) time periods to assess for temporal changes in outcomes. Overall survival (OS) was calculated using the Kaplan–Meier survival function, while non-relapse mortality (death from any cause in the absence of relapse of underlying disease, NRM) was calculated using cumulative incidence probability. Overall survival between the transplant types was compared using the log-rank statistical test, while Gray’s test was used to compare cumulative incidence rates.

## 3. Results

Of 6928 HLA-matched transplants undertaken at 15 Canadian centers, there were 3386 MRD and 3542 MUD allografts. Comparing the two time periods, the frequency of MRD HCTs dropped: there were 1861 MRD HCTs in the early period (54.6% of allogeneic HCTs), and 1525 in the later period (36.6% of allogeneic HCTs). In contrast, MUD HCTs increased from 1197 in the early period (35.1% of allogeneic HCTs) to 2345 in the later period (56.3% of allogeneic HCTs).

### 3.1. Outcomes for Patients with Malignant Diseases

Of the 6928 MRD and MUD allogeneic HCTs undertaken, 6124 (88.4%) were for malignant disease indications. In the early period, the 5-year OS favored MRD HCT (54%, 95% Confidence Interval [CI] 52–57%) over MUD HCT (47%, 95% CI, 43–50%, *p* < 0.001). In contrast, outcomes between donor sources in the later era were comparable, with a 5-year OS of 49% (95% CI 45–53%) for MRD, and 50% (95% CI 47–52%) for MUD HCT, *p* = 0.969. These outcomes are demonstrated in Table 1.

NRM at 2 years for MRD recipients in the early era was 17% (95% CI 15–19%), which was superior to the NRM for MUD recipients of 25% (95% CI, 22–28%, *p* < 0.001). In the later period, the 2-year NRM for MRD was 17% (95% CI, 14–19%) compared to 22% for MUD HCT (95% CI 20–24%, *p* = 0.001), as shown in Table 1.

### 3.2. Outcomes for Patients with Non-Malignant Diseases

Of the MRD and MUD allogeneic HCTs undertaken, 804 (11.6%) HCTs were for non-malignant diseases. As shown in Table 2, the 5-year OS in the early period favored MRD over MUD: 85% (95% CI 79–90%) vs. 63% (95% CI 51% vs. 73%, *p* < 0.001). This difference in favor of MRD was not appreciable in the later era, where MUD HCT recipients experienced similar benefits to MRD HCT: 5-year OS 91% (95% CI 85–94%) vs. 85% (95% CI 77–90%, *p* = 0.209).

The 2-year NRM in the early period for MRD recipients was 11%, (95% CI 7–16%], and was superior to MUD recipients (29%, 95% CI 20–30, *p* < 0.001). In the later era, the 2-year NRM for MRD was 8% (95% CI 4–13%), compared to 15% (95% CI 10–22%) for MUD (*p* = 0.138), as shown in Table 2.

## 4. Discussion

As MUD has become the most frequent donor source for allogeneic HCT in Canada, overall survival for recipients has improved, now matching those for MRD allogeneic HCT. The favorable outcomes after MUD allogeneic HCT were noteworthy in patients with malignant diseases (who comprise most HCT recipients) and in patients with non-malignant diseases.

Improving outcomes for MUD HCT can be explained by several factors. Widespread use of high-resolution HLA typing of class I and II HLA alleles has led to better tissue matching between donors and patients in turn leading to reduced incidence of graft rejection and GVHD. During this time, there have also been innovations in conditioning regimens, as well as in GVHD prophylaxis and treatment, in turn, leading to reduced NRM. Improved patient selection also leads to better outcomes, whereby patients who may not benefit from allografting are advised against this high-risk procedure.

Canadian family sizes are smaller than historical trends. Consequently, the probability of identifying a MRD is lower than 2 to 3 decades ago [2]. Moreover, the age at which patients receive HCT is progressively higher [8], such that when MRDs are identified, the donors themselves may be older. The ability of a family donor to safely donate hematopoietic cells may be limited by the presence of donor comorbidities such as cardiovascular disease or cancer, which are associated with older age [9]. North American and European data support that when faced with a choice between an older MRD (≥50 years) and a younger (≤35–40 years) MUD, the latter group is associated with a reduced relapse rate and improved leukemia-free-survival (LFS) in patients with myeloid malignancies and ALL [5,6,7]. These data highlight the increasing importance of MUDs in the modern practice of HCT.

With respect to patients with non-malignant diseases, we noted convergence in outcomes between MRD and MUD HCT in the later era. However, in patients with non-malignant diseases, the presence of GVHD is associated with adverse clinical effects that are not counterbalanced by graft-versus-malignancy effects. In this context, MRDs, when available, remain the primary donor choice for non-malignant diseases such as aplastic anemia and sickle cell disease. Nonetheless, it is heartening to observe that the frequency of MUD allografts increased in the later era, as did the overall survival in this group. This suggests that access to curative-intent allogeneic transplantation for non-malignant diseases has improved with greater confidence in the availability and safety of MUD HCT. The wider use of reduced condition regimens and augmented immune modulation with anti-thymocyte globulin (ATG), post-transplantation cyclophosphamide (PTCY), and low-dose TBI are all likely to have contributed to the greater uptake and improved outcomes after MUD HCT for non-malignant diseases.

Recently, haploidentical-related and HLA-mismatched unrelated donor transplants have become a practical and emerging possibility for patients without HLA-matched donors. This is especially relevant for patients whose ethnicities (and thus HLA haplotypes) are not well represented in volunteer donor registries [10,11,12]. The current study did not include these alternative donor types, and we acknowledge that the clinical outcomes of patients who received haploidentical-related and mismatched unrelated donor transplants in Canada will also need to be scrutinized.

Amongst MRD recipients, outcomes did not improve over time to the same degree that we observed for MUD recipients. Comparing the outcomes between two transplant eras is limited by the possibility of confounders, especially selection bias: MRD recipients in the earlier era were younger and thus more likely to withstand the risks of transplantation. In addition, they were also likely to have had an inherently better disease prognosis, such as patients with early phase chronic myeloid leukemia (CML) and non-Hodgkin lymphoma, as demonstrated in our initial publication of this cohort [8]. Such patients may have successfully received effective alternative non-transplant therapy in the later era.

In the later era, the 2-year NRM for MUD recipients with malignant diseases remained inferior to that of MRD recipients. It should be noted that the widespread introduction of effective immunoprophylaxis with ATG in MUD HCT occurred late in the second era of this study, as the seminal publication to support its evidence in the Canadian context was in 2016 [13]. Similarly, uptake of PTCY for both matched and mismatched unrelated donors has occurred only recently [12,14]. Consequently, further time would likely be needed to see whether the introduction of ATG and PTCY for MUD HCT confers further reductions in NRM.

A limitation inherent in this cohort study is the possibility of confounding variables that render inter-group comparisons challenging. However, as our primary objective was to compare contemporaneous groups (MRD vs. MUD in the early era and MRD vs. MUD in the later era), we believe that in Canadian clinical practice, the MRD and MUD groups would be reasonably well balanced. Future studies planned by the CTTC Registry will undertake efforts to control for known biases and confounders.

Another limitation is with respect to any differences in adverse events after MRD or MUD HCT. The current analysis examined only lethal toxicities, as reflected by 2-year non-relapse mortality. Future studies planned by the CTTC Registry will undertake efforts to examine other short- and long-term effects such as GVHD and second malignancies. In conclusion, we show that clinical outcomes after MUD allogeneic HCT in Canada now match those of MRD. This is especially noteworthy for patients with malignant diseases who constitute the majority of HCT recipients in Canada. As most HCT candidates are not assured of an available MRD, unrelated donors represent an increasingly important source of cells for Canadian patients.

## Figures and Tables

**Table 1 curroncol-32-00010-t001:** Overall survival and non-relapse mortality for malignant diseases by year of transplant.

	MRD			MUD			
	5-Year OS%						
	No. at Risk	OS%	95% CI	No. at Risk	OS%	95% CI	*p*-Value
2000–2009	538	54	52–57	245	47	43–50	<0.001
2010–2019	184	49	45–53	242	50	47–52	0.969
	2-year NRM %						
2000–2009	738	17	15–19	378	25	22–28	<0.001
2010–2019	402	17	14–19	570	22	20–24	<0.001

Legend: MRD: Matched Related Donor; MUD: Matched Unrelated Donor; OS: Overall Survival; NRM: Non-relapse mortality.

**Table 2 curroncol-32-00010-t002:** Overall survival and non-relapse mortality for non-malignant diseases by year of transplant.

	MRD			MUD			
	5-Year OS%						
	No. at Risk	OS	95% CI	No. at Risk	OS	95% CI	*p*-Value
2000–2009	94	85	79–90	26	63	51–73	<0.001
2010–2019	34	91	85–94	26	85	77–90	0.209
	2-year NRM%						
2000–2009	134	11	7–16	42	29	20–30	<0.001
2010–2019	138	8	4–13	76	15	10–22	0.138

Legend: MRD: Matched Related Donor; MUD: Matched Unrelated Donor; OS: Overall Survival; NRM: Non-relapse mortality.

## Data Availability

The data that support the findings of this study are not publicly available to ensure and maintain the privacy and confidentiality of individuals’ health information. Requests for data may be made to the appropriate data stewards (CancerCare Manitoba’s Research and Resource Impact Committee) who may be contacted via the corresponding author.

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
