# Peer review of "Matched Unrelated Donor Hematopoietic Cell Transplantation: Increased Usage and Improvements in Clinical Outcomes in Canada"

_curroncol, 2024, doi:10.3390/curroncol32010010_

Round 1
Reviewer 1 Report
Comments and Suggestions for Authors
Summary
The study investigates the trends in the usage and outcomes of matched unrelated donor (MUD) hematopoietic cell transplantation (HCT) in Canada compared to matched related donor (MRD) HCT from 2000 to 2019. This study aims to provide insights into the increasing selection of MUDs as donor source for patients lacking suitable MRDs.
General
The manuscript presents a well-structured analysis of the trends and outcomes associated with matched unrelated donor (MUD) hematopoietic cell transplantation (HCT) in Canada. The study is based on a substantial dataset and the statistical methods employed are appropriate. The authors appropriately discuss the implications of their findings in the context of existing literature. Overall, the manuscript provides valuable insights to the field of hematopoietic cell transplantation and is suitable for publication with minor revisions/correction (see below).
Results
Lines 78,79: “Of 7571 first allogeneic HCTs between 2000 and 2019 undertaken at 15 Canadian transplant centres between 2000 and 2019, there were 3386 MRD and 3542 MUD allografts.”
- ‘between 2000 and 2019’ is duplicate
- The numbers don’t sum up: 3386+3542 != 7571
- Later in the text the number is (correctly) given as 6928 = 3386+3542
Author Response
Reviewer 1:
Summary
The study investigates the trends in the usage and outcomes of matched unrelated donor (MUD) hematopoietic cell transplantation (HCT) in Canada compared to matched related donor (MRD) HCT from 2000 to 2019. This study aims to provide insights into the increasing selection of MUDs as donor source for patients lacking suitable MRDs.
General
The manuscript presents a well-structured analysis of the trends and outcomes associated with matched unrelated donor (MUD) hematopoietic cell transplantation (HCT) in Canada. The study is based on a substantial dataset and the statistical methods employed are appropriate. The authors appropriately discuss the implications of their findings in the context of existing literature. Overall, the manuscript provides valuable insights to the field of hematopoietic cell transplantation and is suitable for publication with minor revisions/correction (see below).
Results
Lines 78,79: “Of 7571 first allogeneic HCTs between 2000 and 2019 undertaken at 15 Canadian transplant centres between 2000 and 2019, there were 3386 MRD and 3542 MUD allografts.”
- ‘between 2000 and 2019’ is duplicate
- The numbers don’t sum up: 3386+3542 != 7571
- Later in the text the number is (correctly) given as 6928 = 3386+3542
Author Response
Thanks for pointing out these deficiencies: The relevant sentence now reads:
“ Of 6928 HLA matched transplants undertaken at 15 Canadian centres, there were 3386 MRD and 3542 MUD allografts”
Reviewer 2 Report
Comments and Suggestions for Authors
Seftel, et al. investigated temporal trends in matched unrelated donor (MUD) compared to matched unrelated donor (MRD) HCT from 2000-2019 using national registry data in Canada. The authors found that the proportion of MUD HCTs increased from 35.1% to 56.3% in the early (2000-2009) and later (2010-2019) eras, respectively. Also, the authors found that 5-year overall survival (OS) after MUD HCT for patients with malignant diseases was inferior to MRD HCT in the early era but OS was comparable for the two donor sources. In addition, for patients with non-malignant diseases, 5-year OS after MUD HCT was inferior to MRD in the early era, but in the later era 5-year OS was similar between two donor sources. Therefore, the authors suggested that MUDs are the most common donor source in Canada. This study was well conducted and the results are reasonable and clearly presented. Below are my comments.
Major points 1. Compared to the outcomes of MUD recipients for malignant diseases, MRD HCTs are not improved over time in terms of OS or NRM. Although the authors addressed some reasons for this, other reasons such as GVHD prophylaxis, age of recipients, conditioning regimens, comorbidity index, and risk for relapse should be discussed in depth. 2. Outcomes of MUD recipients for malignant diseases have not dramatically changed. Similar outcomes were reported for MUD and MRD recipients. One reason for this it may due to decreased OS in MRD HCTs. However, the authors did not discuss this. This point is important to understand the current situation as well as to improve outcomes in the future. 3. In contrast to the outcomes for malignant diseases, OS of MUD HCTs for non-malignant diseases dramatically increased and NRM decreased in comparison to MRD HCTs in the current era. The authors should discuss potential reasons for these findings in depth.Author Response
Reviewer 2:
Seftel, et al. investigated temporal trends in matched unrelated donor (MUD) compared to matched unrelated donor (MRD) HCT from 2000-2019 using national registry data in Canada. The authors found that the proportion of MUD HCTs increased from 35.1% to 56.3% in the early (2000-2009) and later (2010-2019) eras, respectively. Also, the authors found that 5-year overall survival (OS) after MUD HCT for patients with malignant diseases was inferior to MRD HCT in the early era but OS was comparable for the two donor sources. In addition, for patients with non-malignant diseases, 5-year OS after MUD HCT was inferior to MRD in the early era, but in the later era 5-year OS was similar between two donor sources. Therefore, the authors suggested that MUDs are the most common donor source in Canada. This study was well conducted and the results are reasonable and clearly presented. Below are my comments.
Major points 1. Compared to the outcomes of MUD recipients for malignant diseases, MRD HCTs are not improved over time in terms of OS or NRM. Although the authors addressed some reasons for this, other reasons such as GVHD prophylaxis, age of recipients, conditioning regimens, comorbidity index, and risk for relapse should be discussed in depth. 2. Outcomes of MUD recipients for malignant diseases have not dramatically changed. Similar outcomes were reported for MUD and MRD recipients. One reason for this it may due to decreased OS in MRD HCTs. However, the authors did not discuss this. This point is important to understand the current situation as well as to improve outcomes in the future. 3. In contrast to the outcomes for malignant diseases, OS of MUD HCTs for non-malignant diseases dramatically increased and NRM decreased in comparison to MRD HCTs in the current era. The authors should discuss potential reasons for these findings in depth.
Author Response
Thanks for pointing these 3 deficiencies. We agree they should be addressed. We have addressed them in combination in the discussion section, in particular these two sections:
“With respect to patients with non-malignant diseases, we noted convergence in outcomes between MRD and MUD HCT in the later era. However, in patients with non-malignant diseases the presence of GVHD is associated with adverse clinical effects that are not counterbalanced by graft-vs-malignancy effects. In this context, MRDs, when available, remain the primary donor choice for non-malignant diseases such as aplastic anemia and sickle cell disease. Nonetheless, it is heartening to observe that the frequency of MUD allografts increased in the later era, as did the overall survival in this group. This suggests that access to curative intent allogeneic transplantation has improved with greater confidence in the availability and safety of MUD HCT. The wider use of reduced conditioning regimens and augmented immune modulation with anti-thymocyte globulin (ATG), post-transplantation cyclophosphamide (PTCY) and low dose TBI have all likely contributed to the greater uptake and improved outcomes after MUD HCT for non-malignant diseases”
“Amongst MRD recipients, outcomes did not improve over time to the same degree that we observed for MUD recipients. Comparing the outcomes between two transplant eras is limited by the possibility of confounders, especially selection bias: MRD recipients in the earlier era were younger and thus more likely to withstand the risks of transplantation. In addition, they were also likely to have had an inherently better disease prognosis, such as patients with early phase chronic myeloid leukemia (CML) and non-Hodgkin Lymphoma, as demonstrated in our initial publication of this cohort8. Such patients may have successfully received effective alternative non-transplant therapy in the later era.”
Reviewer 3 Report
Comments and Suggestions for Authors
In the present report, authors compared the effectiveness of MRD and MUD on malignant and non-malignant diseases. They concluded that MUD and MRD have comparable outcome in HCT treatment. Few comments are:
1. Whether did authors consider age, gender, types of diseases when they compare MRD and MUD during HCT?
2. For the outcomes, whether did authors think of side effects of MRD and MUD during HCT?
Author Response
Reviewer 3
In the present report, authors compared the effectiveness of MRD and MUD on malignant and non-malignant diseases. They concluded that MUD and MRD have comparable outcome in HCT treatment. Few comments are:
- Whether did authors consider age, gender, types of diseases when they compare MRD and MUD during HCT?
Author response:
We thank this reviewer for highlighting the importance of possible confounding variables that may explain differences in outcomes between MRD and MUD allografts.
Our primary paper, published in Current Oncology (Hematopoietic Cell Transplantation Trends and Outcomes in Canada: A Registry-Based Cohort Study), is listed as reference # 8 in the manuscript. In that original paper, we summarize the clinical and demographic characteristics of the entire allograft recipient cohort, comparing the early and late eras. We demonstrate that there were indeed measurable differences in age, cell source, and disease indication between the earlier and later eras, which likely explains some of the early vs. later era outcome differences.
With respect to the possibility of confounding variables that might affect the comparison between MRD vs. MUD outcomes, we agree that one limitation of our current study is that we did not undertake a multivariable analysis in order to control for these possible confounders. However, In the current study, we are comparing MRD vs MUD outcomes in the same era, and for this reason we believe that differences on the above-mentioned confounders are not likely to explain our observed results. As we cannot clearly prove this, we highlight his in the discussion section as follows:
“A limitation inherent in this cohort study is the possibility of confounding variables that render inter-group comparisons challenging. However, as our primary objective was to compare contemporaneous groups (MRD vs. MUD in the early era and MRD vs MUD in the later era) we believe that in Canadian clinical practice the MRD and MUD groups would be reasonably well balanced. Future studies planned by the CTTC Registry will undertake efforts to control for known biases and confounders”.
- For the outcomes, whether did authors think of side effects of MRD and MUD during HCT?
Author response:
The purpose of the current study was to explore major clinical differences between MRD and MUD transplants in Canada. However, we agree with the reviewer that short and long-term toxicities should ideally be part of these outcomes. Indeed, we demonstrate 2-year cumulative non-relapse mortality for all groups one of our key outcomes. Other toxicities were not listed in this study, and we agree that our future work should include outcomes such as graft-versus-host-disease, and second malignancies. We have added this in the discussion section:
"Another limitation is with respect to any differences in adverse events after MRD or MUD HCT. The current analysis examined only lethal toxicities, as reflected by 2-tear non-relapse mortality. . Future studies planned by the CTTC Registry will undertake efforts to examine other short and term effects such as graft-versus-host-disease and second malignancies"